# Effectiveness of mobile phone text message reminder interventions to improve adherence to antiretroviral therapy among adolescents living with HIV: A systematic review and meta-analysis

**Nishant Mehra**[1]*, **Abayneh Tunje**[1,2], **Inger Kristensson Hallström**[1], **Degu Jerene**[3]

1 Child and Family Health Unit, Department of Health Science, Faculty of Medicine, Lund University, Lund, Sweden, 2 School of Public Health, College of Medicine and Health Sciences, Arba Minch University, Arba Minch, Ethiopia, 3 KNCV Tuberculosis Foundation, The Hague, Netherlands

* drnishantmehra16@gmail.com

**Data Availability Statement:** All relevant data are within the manuscript and its Supporting Information files.

## Abstract

### Background

Poor adherence to antiretroviral therapy in adolescents living with HIV is a global challenge. One of the key strategies to improve adherence is believed to be the use of digital adherence tools. However, evidence is limited in this area. Our objective was to investigate the effectiveness of mobile phone text message reminders in improving ART adherence for adolescents.

### Methods

The preferred reporting item for systematic review and meta-analysis guideline was followed. A literature search was done in five databases (PubMed, Web of Science, Embase, Global Health and Cochrane) in August 2020. Additional searches for studies and grey literature were performed manually. We included studies with quantitative design exploring the effectiveness of text message reminders, targeting adolescents aged 10–19 years. Studies were excluded if the intervention involved phone calls, phone-based applications, or other complex tech services. Mean differences between intervention and standard of care were computed using a random effects model. Subgroup analyses were performed to identify sources of heterogeneity between one-way and two-way text messages.

### Results

Of 2517 study titles screened, seven eligible studies were included in the systematic review. The total number of participants in the included studies was 987, and the study sample varied from 14 to 332. Five studies showed a positive impact of text messaging in improving adherence, while no significant difference was found between the intervention and the control (standard of care) group in the remaining two studies. The pooled mean difference

**Funding:** The study was supported by the Swedish Research Council for Health, Working Life and Welfare Life (FORTE) (https://forte.se/en/) program support 2018–01399 and Swedish Research Council (Vetenskapsrådet) (https://www.vr.se/english.html) program support VR 2016-05706. The funder had no role in the design of the study and collection, analysis, and interpretation of data or in writing the manuscript. The grant was awarded to IKH and DJ.

**Competing interests:** The authors have declared that no competing interests exist.

**Abbreviations:** AIDS, Acquired Immuno-deficiency Syndrome; ART, Anti-retroviral therapy; HIV, Human Immunodeficiency Virus; LMIC, Low- and middle-income countries; MEMS, Medication Event Monitoring System; RCT, Randomised control trial; ROBINS-I, Risk of bias in non-randomised studies of interventions; SMS, Short message service; USA, United States of America; WHO, World Health Organization.

between the intervention and the control group was 0.05 (95% CI: –0.08 to 0.17). There was considerable heterogeneity among the studies ($I^2$ = 78%).

## Conclusion and recommendation

The meta-analysis of text message reminder interventions did not show a statistically significant difference in the improvement of ART adherence among adolescents living with HIV. The included studies were heterogeneous in the reported clinical outcomes, where the effectiveness of the intervention was identified in small studies which had a short follow-up period. Studies with bigger sample size and a longer follow-up period are needed.

## Introduction

Human Immunodeficiency Virus (HIV), which causes Acquired Immunodeficiency Syndrome (AIDS), is one of the world's most serious health and development challenges. Since the beginning of the HIV pandemic, more than 76 million people have been infected with HIV [1], and about 38 million people are currently living with HIV [2]. More than half of the disease burden for HIV is in East and Southern Africa, with an estimated 730,000 new cases in 2019. The latest estimates suggest that approximately 1.7 million adolescents have HIV, accounting for about 5% of all people living with HIV (PLHIV) [3, 4]. The definition of an adolescent used in this study is a person aged 10–19 years [5]. The adolescent age group is further subcategorised into early adolescents (10–15 years), middle adolescents (14–17 years) and late adolescents (16–19 years) [6].

With an emphasis on viral suppression, the Joint United Nations Programme on HIV/AIDS launched a 90-90-90 initiative in 2014 [7]. A target was set for 2020 that 90% of people living with HIV know their status, 90% of those who know their status start treatment, and 90% of those on treatment achieve viral suppression [7]. At the end of 2019, 12.6 million people (33% of people living with HIV) did not have access to antiretroviral therapy (ART) [8]. An estimated 840,000 children did not have access to ART at the end of 2019, amounting to 47% of all children living with HIV [8]. Even though ART has achieved great success, the 2020 treatment targets seem unachievable for many countries. Because of significant improvements in access to ART over the last decade, the annual AIDS-related death rate has declined significantly since 2003, with over 70% decrease in the age range of 0–9 years, but the decline in adolescents is a mere six percentage points [9, 10]. In 2019, global AIDS-related deaths among those aged below 15 years numbered 95,000 [8]. Also, retention in care and adherence to ART regimens have been increasingly seen as a challenge in the global fight against this pandemic. Treatment adherence is defined as how patients' medication-taking behaviour corresponds to their health care professionals' recommendations, including taking medications daily at roughly the same time without missing doses [11]. It is the key to better health outcomes in chronic diseases like HIV, and it also helps to limit onward transmission, thus helping to reduce the burden of disease [10, 12]. For optimal viral suppression, an adherence rate of more than 95% is widely cited [13]. However, recent studies suggest that even moderate adherence to potent ART regimens can achieve viral suppression [14]. Parienti et al. point to sustained treatment interruptions leading to viral rebound rather than interspersed missed doses [15]. Studies suggest that globally about 60% of adolescents adhere to ART [16]. Poor ART adherence leads to accelerated progression of PLHIV to AIDS [17], resulting in increased morbidity and mortality [18].

Frequently reported barriers to adherence to ART among adolescents are forgetfulness [19–21], stigmatisation of status disclosure [19, 21, 22], adverse effects of medication [19], anxiety, depression and substance abuse [19–22], financial constraints [21] and accessibility issues [19]. For these reasons, the World Health Organization (WHO) asserts the need for a better support system for adolescents to improve their adherence to ART [23].

Since the beginning of the 21st century, access to the internet and mobile phones among adolescents has improved globally [24]. This has stimulated an interest in exploring the role of information, communication, and technology in health, also known as eHealth [25]. The WHO Global Observatory for eHealth defines eHealth as "the use of information and communication technologies (ICT) for health" [26]. This involves the delivery of health information by using mobile phones' short message service (SMS), patient monitoring devices, and internet-based components (social media, computer software, websites, mobile apps, games, and chat rooms) [27]. Mobile health (mHealth) technology is a specific kind of eHealth service that is defined by the Healthcare Information and Management Systems Society (HIMSS) as the use of small, portable computers or telecommunication equipment to meet the needs of health care and improve the quality of clinical research and healthcare on a global scale.

In the past decade, several studies have explored the acceptability, feasibility, and utility of eHealth and mHealth interventions to improve adherence among patients living with chronic diseases [28–31]. The advantages of using mHealth interventions are their low cost, making them suitable for use in low- and middle-income countries (LMIC), along with their convenience and accessibility. Besides, mHealth can provide users with a private space to curb the discrimination and stigma associated with HIV [32–35]. However, the path to mHealth incorporation into clinical care is fraught with many challenges [36].

In this study, the effect of a text message reminder intervention on ART adherence in adolescents living with HIV is explored by performing a systematic review and a meta-analysis. A text message is a simple mHealth service where patients receive an SMS reminding them to take their medication on time. The SMS is sent at a predetermined time, usually before the scheduled medication intake time for the patient as recommended by their doctors. Commonly the service can be categorised as either a one-way text message (patient receiving an SMS from the care provider) or a two-way text message (patient having the opportunity to respond to the SMS). A text message can be received on a mobile phone with basic features. This intervention avoids stigma and can be rolled out in a resource-limited setting. Previous studies have shown the effectiveness of mHealth interventions in improving treatment adherence among adults living with HIV [37–40], and the evidence is emerging in the adolescent age group.

This study therefore aims to identify and investigate the current literature looking into the effectiveness of mobile phone text message reminders in improving ART adherence among adolescents.

## Methods

The preferred reporting item for systematic review and meta-analysis (PRISMA) guidelines was followed [41]. The search was based on a PICO (Population/Problem, Intervention, Control and Outcome) question with explicit inclusion and exclusion criteria: Does targeted mobile text messaging improve adherence to antiretroviral treatment among adolescents living with HIV compared to a "standard of care"?

### Criteria for study selection

The inclusion criteria were: all peer-reviewed articles, registered trials and conference abstracts and posters available in English where the search for the records was not restricted to a time

limit; HIV as the primary focus of the study; mobile phone text messaging (SMS) reminder as an intervention to improve adherence; using either adherence to ART (measured in different ways) or viral load as the study outcome; quantitative study design investigating the impact of text message reminders (cohort studies and randomised control trials (RCT)).

Studies were excluded if: phone calls or sophisticated mHealth or eHealth interventions such as smartphone applications, social media, internet etc.; the majority of study participants (more than 50%) do not belong to the adolescent age group as ascertained by reported mean age and standard deviation; qualitative study design, systematic reviews and meta-analysis and quantitative study design that do not account for effectiveness (cross-sectional studies etc.).

## Source of data and search strategy

Under the guidance of a librarian, a comprehensive search strategy was developed. The search was performed using a combination of keywords and truncated terms in two sets using Boolean search strategy (intervention and disease): "mobile phone*", "cell phone*", "text reminder*", "text messages*", "phone based*" and "HIV", "AIDS", "ART", "Antiretroviral therap*". The literature search was done on PubMed, Embase, Web of Science, Global Health (CABI) and Cochrane databases on 18 August 2020. In the Embase database search, citations unique to this database were only included using the "Source" function. Additional articles were identified by hand searching the reference list of the screened articles, www.clinicaltrials. gov, Grey Literature Report, International AIDS Society–conference abstracts, Conferences on Retroviruses and Opportunistic Infections [42]. Citations of all studies were downloaded to EndNote 8 citation manager software [43], where they were further sorted.

## Study selection

After removing the duplicates from the search hits, first titles and then abstracts were screened by NM based on the inclusion criteria. Later, full texts were read by two authors (NM and AT) and assessed based on the exclusion criteria. Fig 1 shows the detailed process of the screening and selection process. Disagreement concerning the inclusion of a study was resolved by the third and fourth researcher (IKH and DJ).

## Data extraction and quality assessment

For each study, data on demographics, interventions and outcomes were extracted independently by two authors (NM and AT). This process was guided by Cochrane's data collection recommendations [44]. Later data were tabulated into two tables: (1) study characteristics and (2) interventions and adherence results. Cochrane's risk assessment tool was used to assess the quality of the included randomised studies at the level of individual studies [45]. Studies with a non-randomised study design were assessed using the Risk of Bias in Non-randomised Studies of Interventions (ROBINS-I) tool [46].

## Quantitative analysis

A meta-analysis was undertaken where pooled mean differences in adherence for the intervention versus standard of care was calculated. Most studies included in this systematic review reported adherence to ART as a continuous outcome (mean adherence). Therefore, mean adherence was chosen as the outcome measure for quantitative analysis. For the meta-analysis, adherence is defined as how a participants' pill count or MEMS data corresponds to their doctor's prescription regimen during the period of trial. Mean adherence is a continuous variable that can take a value from 0–1, where 0 represents nil adherence, and 1 represent perfect

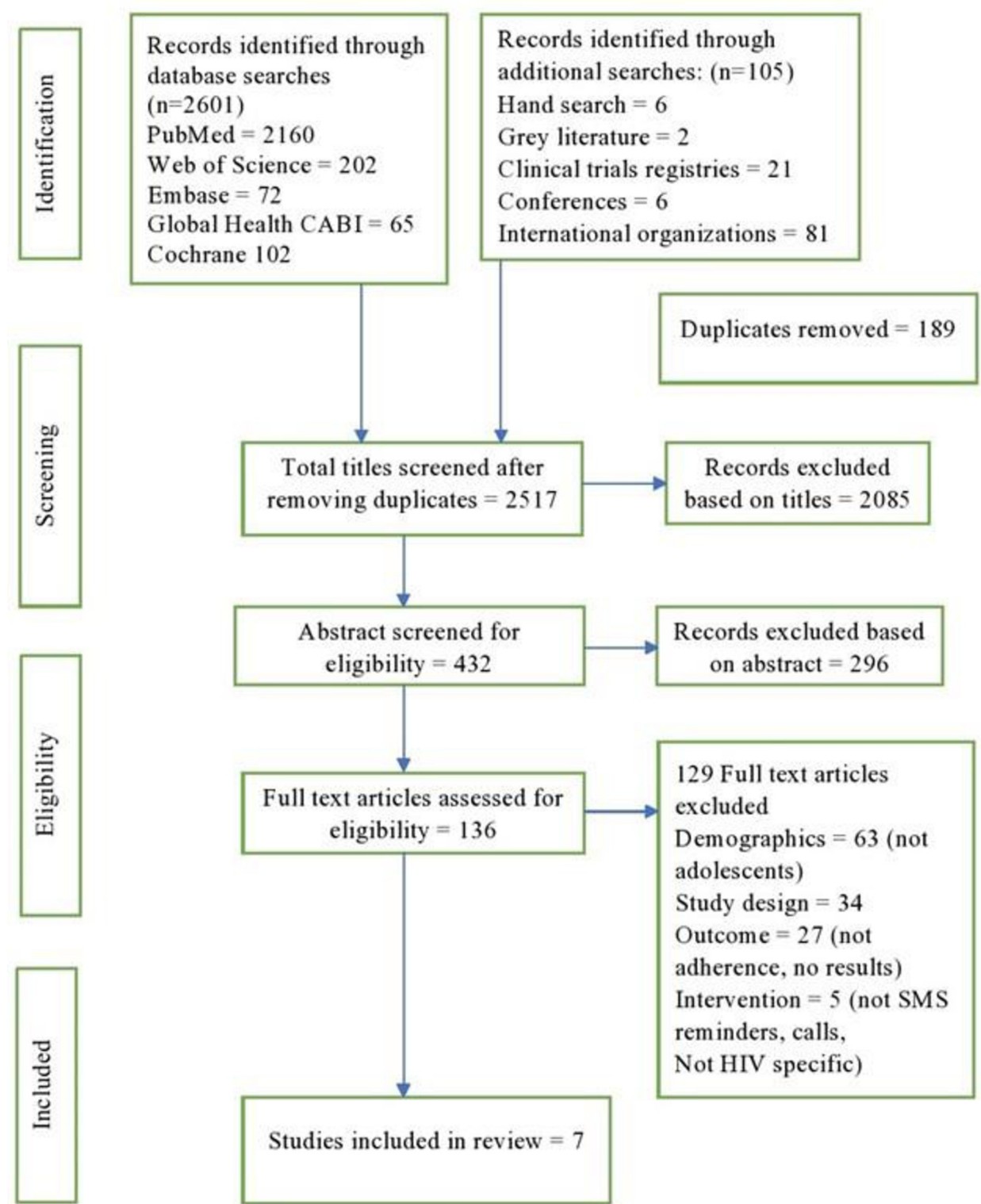

**Fig 1. Flowchart describing studies selection process.**

adherence. The data used in the analysis were mean adherence rates, standard deviation, and the number of participants in each intervention arm. Where standard deviation was not available, other effect measures such as confidence intervals were converted to fit the model. If it was not possible to transform the effect measures without access to raw data, studies were excluded from the meta-analysis. Due to the anticipated heterogeneity of the structure of intervention, a random effects model was used to estimate the pooled mean difference between the intervention arms with a 95% confidence interval. Studies with a similar intervention (one-way text message, two-way text message and short follow-up (<6 months)) were further analysed in sub-groups using a fixed effects model. All statistical analyses were performed using Review Manager (version 5.4; The Nordic Cochrane Centre, The Cochrane Collaboration, Copenhagen, Denmark).

## Results

### Study characteristics

A total of 2517 study titles were screened based on the study selection criteria. Seven studies were included in the review; five RCTs (two in the USA and one each in Uganda, Nigeria, and Cameroon) and two cohort studies (one each in USA and Argentina) analysing the effect of mobile text reminders for improving adherence to ART among adolescents living with HIV were included (Fig 1). The total number of participants in the included studies was 987, and the study sample varied from 14 to 332 (Table 1). Of the seven studies included, four were peer-reviewed articles [47–50], two studies were registered trials with available results [51, 52] and a conference poster with results [53].

The mean age of the participants in most studies was in the late adolescent age group (16–19 years). One study had over 80% male participants [47], another had 69% females [50]. The remaining studies were more gender-equal. Six studies reported sub-optimal baseline adherence levels among recruited participants. Three studies [47, 48, 51] mentioned recruiting patients with experience of treatment (patient started ART at least six months before recruitment to trial) (Table 1). One study mentioned speaking English as a criterion for inclusion [53]. Access to mobile phones was an inclusion criterion in all the included studies. The study participants in all studies received text messages. In the study by Stankievich et al., about 30% of the participants' parents received the text message [50].

### Quality assessment and risk of bias

Of the five RCTs that were analysed using the Cochrane risk assessment tool, three–Linnemayr (2017), Abiodun (2019), and Ketchaji (2019)–were deemed to have a low risk of bias (Fig 2). The main concern with these studies was that the participants in different intervention arms were not blinded due to the unique nature of the intervention. Mimiaga's (2019) study was judged to be at high risk of bias due to incomplete and selective reporting of outcomes compared with their research protocol [49, 55]. Jeffries' (2016) study was judged to have an unclear risk of bias as this was a poster study and raw data could not be accessed. The two cohort studies by Hailey (2013), and Stankievich (2014), were assessed based on the ROBINS-I tool and judged to be at serious risk of bias as they lacked a control group. Overall, studies with a bigger sample size were found to be at low risk of bias [48, 52].

### Summary of included studies

Overall, four studies had only text messages as their intervention, while the others included counselling or peer support as a part of both intervention and standard of care (Table 2).

**Table 1. Characteristics of included studies.**

| Author | Study | Study | Gender | | Age of | ART treatment | Adherence status/ |
|---|---|---|---|---|---|---|---|
| Year | design | size (N) | Male | Female | participants | experience | viral load at the |
| Country | | | | | (years) | | time of recruitment |
| **Hailey** [47] | Cohort | 87 | 70 | 17 | 15–24 | New and existing | Sub-optimal adherence |
| 2013 | | | | | | patients | |
| USA | | | | | | | |
| **Stankievich** [50] | Cohort | 22 | 7 | 15 | 6–25 | Unspecified | Viral Load |
| 2014 | | | | | mean 17 | | >1000 copies/ml |
| Argentina | | | | | | | |
| **Jeffries** [53] | RCT | 136 | NA | NA | 15–24 | >1 year– 60% | New patients or |
| 2016 | | | | | | <6 months– 24% | sub-optimal adherence |
| USA | | | | | | | |
| **Linnemayr** [48] | RCT | 332 | 151 | 181 | 15–22 | Treatment experienced | Not reported |
| 2017 | | | | | mean 18 | | |
| Uganda | | | | | | | |
| **Abiodun** [52] | RCT | 212 | 108 | 104 | 15–19 | Unspecified | Sub-optimal adherence |
| 2019 | | | | | mean 16 | | |
| Nigeria | | | | | | | |
| **Ketchaji** [54] | RCT | 184 | NA | NA | 15–19 | >6 months | Sub-optimal adherence |
| 2019 | | | | | | | |
| Cameroon | | | | | | | |
| **Mimiaga** [49] | RCT | 14 | 8 | 6 | 16–24 | Unspecified | Sub-optimal adherence |
| 2019 | | | | | mean 19 | | |
| USA | | | | | | | |

RCT–Randomised Control Trial; NA–Not Available

The primary outcome was reported either as medication adherence or viral load. Three studies (Hailey 2013, Abiodun 2019, and Ketchaji 2019) used self-reported data for outcome assessment, while two studies (Linnemayr 2014 and Mimiaga 2019) documented using a medication event monitoring system (MEMS) in addition to self-reported data. Medication adherence was reported as a mean adherence rate, odds ratio, or using a visual analogue scale (VAS) score. A laboratory test was used to report viral load, and this was reported as number of copies/ml. The follow-up period ranged from 4 months to 24 months. Most studies had a drop-out rate of less than 5% [47, 49, 52], while two studies had a drop-out of up to 19% [48, 54].

## Assessment of the effectiveness of text messaging

Of the total seven reviews, five reported a positive impact of text messaging in improving treatment adherence: 3 RCTs (Jeffries 2016, Ketchaji 2019, and Mimiaga 2019) and 2 cohort studies (Hailey 2013 and Stankievich 2014) (Table 2). RCTs by Linnemayr (2014) and Abiodun (2019) showed no significant difference between the comparison groups. The unit of analysis in the meta-analysis was mean adherence, and this was the reported effect measure in the 3 RCTs: Abiodun et al. (2019), Linnemayr et al. (2014), and Mimiaga et al. (2019).

The pooled mean adherence difference was found to be 0.05 (Fig 3). However, these studies showed statistical heterogeneity ($I^2 = 78\%$). Sub-group analysis of studies found no difference in mean ART adherence (mean difference = 0.00) among adolescents who receive only text

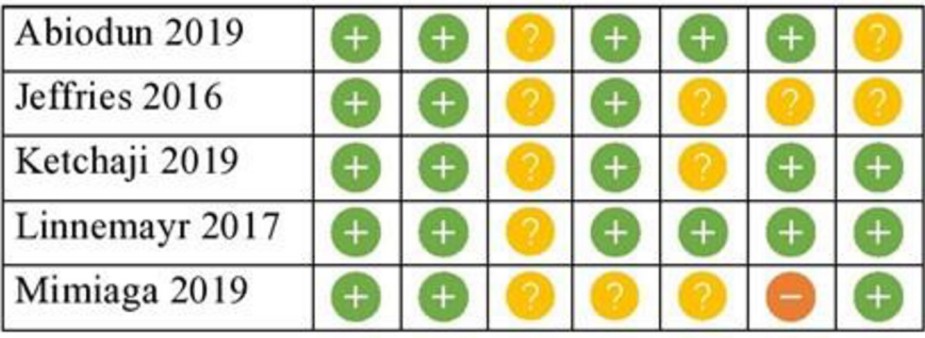

**Fig 2. Quality assessment of included studies that were randomised.**

message (one way) (Fig 4). For a sub-group analysis of a two-way text message intervention, the pooled mean difference was –0.03 (Fig 5).

A third subgroup analysis focusing on studies with a short follow-up period suggests that the overall effect of the intervention is insignificant (p-value = 0.15) (Fig 6).

## Discussion

This systematic review identified seven eligible studies, which consisted of five RCTs and two cohort studies. The cohort studies showed a positive effect of the intervention but were judged to be at serious risk of bias as they lacked a control group. Of the five RCTs, three [47, 49, 51] demonstrated the effectiveness of mobile text messaging in improving ART treatment adherence among adolescents. However, the remaining two RCTs [48, 52] did not significantly differ between the intervention and comparison groups. A meta-analysis of three studies [48, 49, 52] showed considerable heterogeneity among the included studies and no significant pooled mean difference between the intervention arms. The meta-analysis findings were further affirmed by sub-group analyses, where no significant effect of the intervention was found compared to the standard of care.

To our knowledge, this is the first adolescent-specific systematic review of the effectiveness of mobile text messaging in improving adherence to ART. Several previous reviews assessing the effectiveness of text message reminders and other mobile phone-based interventions in improving ART adherence found the intervention to be effective among adults [37, 38, 56]. A meta-analysis study performed in the USA among adults revealed that mobile text message reminders significantly improved antiretroviral therapy adherence and viral load and/or CD4 +count [57]. This was not evident in the present meta-analysis, which may be attributed to the

**Table 2. Reviewed studies.**

| Author | Intervention | Follow-up | Drop-out rate | Outcome measures | Adherence and/or viral load measurement[1] |
|---|---|---|---|---|---|
| Year | (no. of participants) | period | | Measurement tool | |
| Country | | | | Unit | |
| Hailey | **I**: Phase one: 2-way SMS (86) | 24 months | Drop-out 0% | ART adherence | Baseline = 40–50% |
| 2013 | Phase two: video DOT (1) | | | | |
| USA | | | | Self-reported | 24 months = 80% |
| | **C**: No control group | | | | |
| | | | | Mean score | |
| Stankievich | **I**: Mobile generic contact | 32 weeks | Drop-out 9% | Viral load | Baseline 1020–500,000 |
| 2014 | Twice a month text message | | | | (mean 25,100) |
| Argentina | and Facebook (22) | | | Blood test | |
| | **C**: No control group | | | | 32 weeks |
| | | | | mean copies/ml | Undetectable in 65% |
| | | | | proportion of copies/ml | <1000 in 70% |
| Jeffries | **I**: 12 tailored messages | 6 months | Drop-out not | Viral load | Baseline |
| 2016 | per week (91) | | reported | | I: 2.08, C: 2.28 (0.64) |
| USA | | | | Blood test | |
| | **C**: SOC (45) | | | | 3 months |
| | | | | unit (p-value) | I: 1.96, C: 3.5 (0.039) |
| | | | | | 6 months |
| | | | | | I: 1.5, C: 3.94 (0.003) |
| Linnemayr 2017 Uganda | **I1**: one-way SMS (110) **I2**: two-way SMS (110) **C**: SOC (112) | 48 weeks | Drop-out 18% | ART adherence Self-reported mean score at baseline MEMS at 48 weeks mean (95% CI) p-value | Baseline |
| | | | | | I1: 81%, I2: 81%, C:81% |
| | | | | | 48 weeks |
| | | | | | I1: 0.64 (0.58, 0.70) |
| | | | | | I2: 0.61 (0.56, 0.67) |
| | | | | | Pooled I: 0.63 (0.59, 0.67) |
| | | | | | Control: 0.67 (0.62, 0.72) |
| | | | | | p-value 0.35 |
| Abiodun | **I**: SMS with counselling/ group | 20 weeks | Drop-out 1% | ART adherence | Baseline |
| 2019 | chat (106) | | | | I: 0, C: 0 |
| Nigeria | | | | Self-reported | |
| | **C**: counselling/group | | | | 20 weeks |
| | chat (106) | | | VAS score > = 95% | I: 57(54.3%), C: 47(45.2%) |
| | | | | | |
| | | | | | Baseline NA |
| Ketchaji | **I1**: Peer support (46) | 6 months | Drop-out 19% | ART adherence and viral load | 6 months |
| 2019 | **I2**: Daily SMS (46) | | | | Self-reported and pill count |
| Cameroon | **C1**: SOC (46) | | | Self-reported and pill count | I1: 4.1 (1.6–10.9) |
| | **C2**: SOC (46) | | | Blood test | I2: 5.8 (2.3–14.9) |
| | | | | | |
| | | | | Odds ratio (95%CI) | Viral load suppression |
| | | | | | I1: 14.7 (4.8–44.6) |
| | | | | | I2: 15.6 (4.2–57.7) |
| Mimiaga | **I**: Step 1: daily 2-way message | 4 months | Drop-out 0% | ART adherence | Baseline |

*(Continued)*

**Table 2.** (Continued)

| Author | Intervention | Follow-up | Drop-out rate | Outcome measures | Adherence and/or viral load measurement[1] |
|---|---|---|---|---|---|
| Year | (no. of participants) | period | | Measurement tool | |
| Country | | | | Unit | |
| 2019 | Step 2: five counselling | | | | score 74% (SD = 35.3) |
| USA | sessions (7) | | | MEMS and self-reported | |
| | | | | | 4 months |
| | C: SOC (7) | | | Mean score (SD) | I: mean change score |
| | | | | | 13% (SD = 29.5) |
| | | | | | C: mean change score |
| | | | | | −26% (SD = 26.0) |

I–Intervention; I1 –First intervention; I2 –Second intervention; C–Control; SOC–Standard of Care; MEMS = Medication Event Monitoring System; SMS = Short Message Service, NA–Not Applicable

1 –Adherence and viral load measured at baseline and at the end of the follow-up period.

heterogeneity of the studies included. However, a sub-group analysis for a one-way text message with no statistical heterogeneity also lacked an effect in improving adherence.

Studies in the adult age group in LMIC countries showed improved adherence in small samples and short follow-up [58, 59]. In the present review, findings from LMIC are unclear as the study from Cameroon [54] found improved adherence among adolescents, but this was contrary to the findings from studies conducted in Nigeria [52] and Uganda [48]. Most studies with a positive impact of text messaging in the current review were from high-income countries and had a small sample size [47, 49, 50]. However, these were judged to be at high risk of bias. Additionally, a small study effect cannot be ruled out in this case. These findings highlight the differences between adults and adolescents in their response to mobile phone text messaging regarding adherence to ART and the quality of evidence currently available. Fewer studies focusing on the adolescent age group may be a reason behind the differing findings.

Most of the studies included in the present review enrolled participants in the late adolescent age group [47–49, 51–53]. This may be because, in most countries, the legal age of consent for medical intervention is in the late adolescent years. Thus, age may be a deterrent for conducting trials with the early adolescent age group because obtaining a guardian's consent may affect privacy. Accessibility to mobile phones was an inclusion criterion in all the studies, and this may have discouraged researchers from enrolling early age adolescents as they are comparatively less likely to have access to a mobile phone. Although adolescents' access to mobile phones has increased in the past couple of decades, it still varies between countries depending on individuals' socioeconomic status and the cultural norms of the society [60]. A recent study showed the acceptability of a text message reminder intervention to patients in urban and

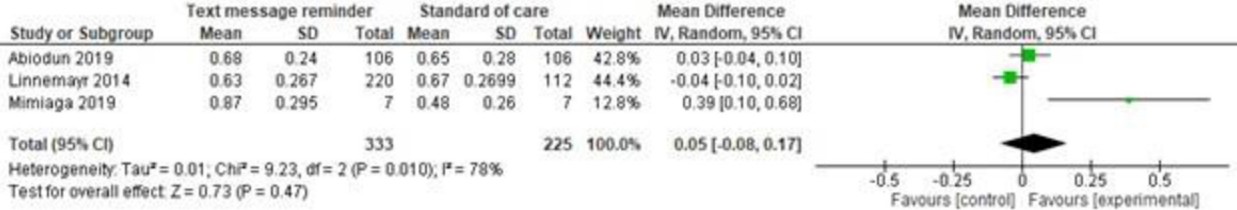

**Fig 3. Forest plot of text message reminder versus standard of care for ART adherence.**

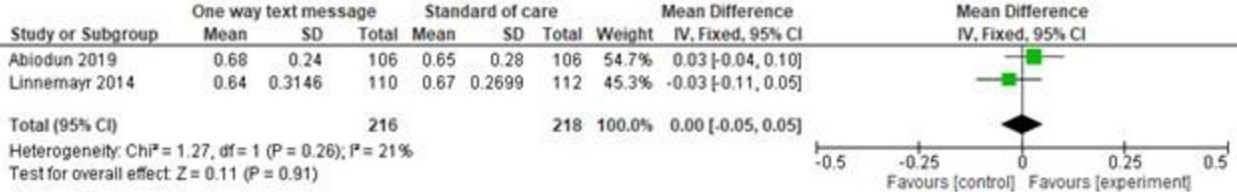

**Fig 4. Forest plot of a sub-group analysis of one-way text message versus standard of care for ART adherence.**

rural settings [60]. However, a study by Rana et al. [61] asserts the need to maintain confidentiality, which can be challenging if participants do not own a mobile phone or share it with other people at home [61]. Therefore, the feasibility of implementing such interventions in adolescents needs to be explored further at a community level.

Though eHealth and mHealth interventions are increasingly being used for chronic health conditions, this review found that few studies have looked at the late adolescent age group (16–19 years). The current evidence base is particularly meagre in resource-limited settings where we expect patients to benefit more from low-cost mHealth interventions. Therefore, more trials with larger study populations in LMIC are called for, and more evidence for the early adolescent age group is required.

Based on the overall assessment, the effectiveness of text reminders in improving adherence to ART remains inconclusive due to small studies that enrolled non-adherent treatment initiators. This might be due to the lower power of the included studies and the fact that few interventional studies have been conducted in the adolescent age group. There is a need to explore the potential of text messaging interventions further, especially with multi-component elements such as two-way messaging in adolescents. Future trials should have a larger sample size and a more extended follow-up period to have more power in the findings. The potential of text messages beyond reminders needs to be further explored.

The present study has both strengths and limitations. This is a comprehensive review of text message reminder interventions to improve ART adherence in the adolescent age group. In the process, a thorough search strategy was implemented under the guidance of an expert librarian, strengthening the validity of our findings. Further attempts were made to reach out to the authors of the included studies to gain access to raw data. Another strength of this study was the use of a mean age of participants in the included studies as a criterion, which contributes to the credibility of our findings as we intended to investigate a specific population. Since adolescent-specific systematic reviews are lacking on this topic, it makes this study a pioneering contribution to the field.

We also acknowledge some limitations in our study. Of the seven studies included in the current review, three were not published in peer-reviewed journals. This raises concerns about the overall quality of the evidence produced by these studies, and therefore entails the limitation that the results are based on data with questionable credibility. Moreover, five studies

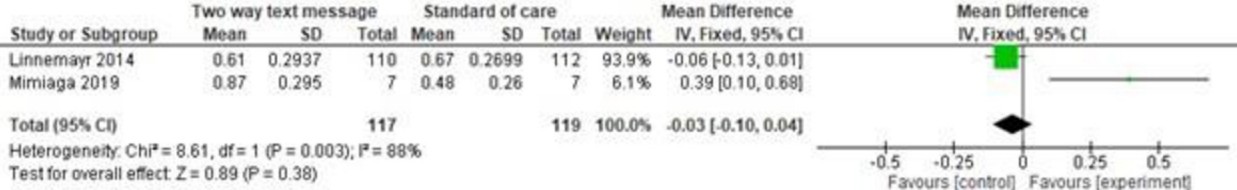

**Fig 5. Forest plot of a sub-group analysis of two-way text message versus standard of care for ART adherence.**

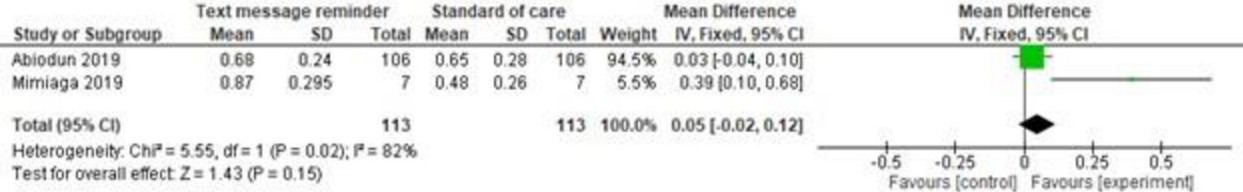

**Fig 6. Forest plot of a sub-group analysis of text message versus standard of care of studies with short follow-up period.**

used self-reported data, which is considered a less sensitive measure of adherence than MEMS and tends to overestimate adherence [62]. We think that the inclusion of self-reported data in our meta-analysis may affect the validity of our findings, and this is therefore a limitation of the present study. Finally, variation in reported outcome (continuous or binary) and units posed a challenge to include most studies in the meta-analysis. Odds ratio is a more common outcome measure for adherence in a meta-analysis that evaluates a binary outcome. In the current study, the included studies dichotomised adherence at different points that posed a challenge to have odds ratio as an outcome measure. Nevertheless, the findings of this meta-analysis can guide the research potential of text messaging in adolescents.

## Conclusion

There is a limited number of studies assessing the effectiveness of mobile phone text reminders in improving adherence to ART among adolescents. Adolescents were the sole recipients of the text messages in most of the studies. The few available studies have critical methodological weaknesses, including a lack of control groups and small sample sizes. Our review of the available evidence nevertheless suggests a positive impact of mobile phone text reminders on adherence to ART among adolescents. The included studies were heterogeneous in the reported clinical outcome. The meta-analysis showed no significant improvement in adherence. Further studies with a more rigorous design are needed to assess the effectiveness of mobile phone text messaging in improving ART adherence in the adolescent age group.

## Supporting information

**S1 Appendix. Search strategy.**
(DOCX)

**S2 Appendix. PRISMA checklist.**
(DOCX)

**S3 Appendix. Data for meta-analysis.**
(XLSX)

## Acknowledgments

The authors would like to thank Ms. Maria Bjorklund for her guidance in developing the literature search strategy.

## Author Contributions

**Conceptualization:** Abayneh Tunje, Inger Kristensson Hallström.

**Data curation:** Nishant Mehra.

**Formal analysis:** Nishant Mehra, Abayneh Tunje.

**Funding acquisition:** Inger Kristensson Hallström, Degu Jerene.

**Investigation:** Nishant Mehra.

**Methodology:** Nishant Mehra, Abayneh Tunje, Degu Jerene.

**Project administration:** Inger Kristensson Hallström, Degu Jerene.

**Supervision:** Inger Kristensson Hallström, Degu Jerene.

**Writing – original draft:** Nishant Mehra, Abayneh Tunje.

**Writing – review & editing:** Nishant Mehra, Abayneh Tunje, Inger Kristensson Hallström, Degu Jerene.

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
