## [Decision Letter · Decision Letter 0]

5 Mar 2021

PONE-D-21-02241

Effectiveness of mobile phone text message reminder interventions to improve adherence to antiretroviral therapy among adolescents living with HIV: a systematic review and meta-analysis.

PLOS ONE

Dear Dr. Mehra,

Thank you for submitting your manuscript to PLOS ONE. After careful consideration, we feel that it has merit but does not fully meet PLOS ONE’s publication criteria as it currently stands. Therefore, we invite you to submit a revised version of the manuscript that addresses the points raised during the review process.

You will see that the Referees found your work of some interest. However, they also raised major criticisms and did not grant your paper enough priority to recommend publication (see coments below). However, if you think all objections raised by the referees can be considered and if additional data requested by reviewers can be provided, we may be willing to reconsider your manuscript.

We look forward to receiving your revised manuscript.

Kind regards,

Giuseppe Vittorio De Socio, MD, PhD

Academic Editor

PLOS ONE

Journal Requirements:

Reviewers' comments:

Reviewer's Responses to Questions

**Comments to the Author**

1. Is the manuscript technically sound, and do the data support the conclusions?

Reviewer #1: Partly

Reviewer #2: Yes

2. Has the statistical analysis been performed appropriately and rigorously? 

Reviewer #1: N/A

Reviewer #2: Yes

3. Have the authors made all data underlying the findings in their manuscript fully available?

Reviewer #1: No

Reviewer #2: Yes

4. Is the manuscript presented in an intelligible fashion and written in standard English?

Reviewer #1: Yes

Reviewer #2: Yes

5. Review Comments to the Author

Reviewer #1: The authors stated that "the data are fully available without restriction" and "all relevant data are within the manuscript and its supporting files". I was not able to find any data files or analysis code. I'd be happy to complete a full review once the data and code are shared. I'm marking as "reject" at this point on the recommendation of the handling editor as a way to request access to these materials. This recommendation is not based on any assessment that the paper is unsuitable for publication.

Reviewer #2: To the authors:

This manuscript addresses an extremely important topic and I applaud the authors for undertaking this systematic review and meta-analysis to help us understand whether mobile phone SMS hold promise to support improved adherence in a known vulnerable population: adolescents with HIV. I enjoyed reading the manuscript. I found a few places that need clarification. With some careful editing, I believe it will be worthy of publication.

Specific Comments:

Abstract

1. In the Results section, 3rd sentence, the authors write, “… while no significant difference was found in the standard of care in the remaining two studies.” This is confusing. Do you mean that no difference was found between intervention and control (standard of care) groups? This should be clarified.

2. In the Conclusion and recommendation section, I think it would be good to highlight more prominently that this study could not confirm its motivating question: whether mobile phone text message interventions can improve adherence among HIV-positive adolescents. The meta-analysis failed to show a significant difference – it’s there in this section but it’s a bit buried between other points. This fundamental finding should be highlighted, I think.

Introduction

3. On page 3, second paragraph, there are some confusing numbers provided. In line 6 of the paragraph (lines 67-68) beginning with “With an emphasis on viral suppression…”, the sentence that says, “… of which 53% were aged 0-14 years and 68% aged 15 and older living with HIV” doesn’t make sense. Can you clarify these numbers?

4. In the same paragraph, some of the references seem a bit old. For instance, the UNAIDS Global AIDS update is cited from 2017 – but the 2020 report is out, so I would recommend using the most recent numbers throughout this important background paragrph.

5. On page 4, in the first paragraph, top line of the page (line 75), the authors write, “Adherence to treatment, defined as how medication-taking behavior corresponds to the recommendations….” is not very helpful as an operational definition. What you mean is more like – “defined as taking HIV medications as prescribed, which includes taking them every day at roughly the same time, without missing doses.”

6. For the last sentence in the same paragraph (line 81), why is the Finitsis article on text message interventions the reference for a general statement on the importance of adherence in preventing progression to AIDS, morbidity, and death among PLHIV? I think it would be good to have a reference to a study whose purpose was to investigate the relationship between adherence and morbidity/death (such as Paterson et al (2000) and Arnsten et al (2001)).

7. On the next page (5), in the last paragraph (line 109), the authors write about their study focusing on the effect of text message reminders to improve ART adherence – but don’t specify that they are also focusing on a specific population: adolescents. This is an important part of their study and should be included here.

Methods

8. In the section on Study inclusion criteria, the text on the adherence measure seems confusing and inaccurate. The text reads (in line 138) “Adherence to ART either as pill count or viral load as an outcome.” One problem with this as written is that, while both outcomes are important, and HIV viral load can be an indicator of past adherence, it is not a measure of adherence itself, strictly speaking. The second problem is that saying “…ART either as pill count or viral load” suggests that adherence will be measured as pill count only. Pill count has a specific meaning in measuring adherence, and it is distinct from such measures as self-report, MEMS, etc. and the latter are clearly included in the papers included in the review – so this is very confusing. I would recommend dropping the “pill count” altogether (as that is misleading) and then revising the line somewhat to read: “using either adherence to ART (measured in various ways) or HIV viral load as a study outcome”. In reading the manuscript, this captures what the authors actually did.

9. For study inclusion/exclusion generally, I would not use bullets. I would recommend writing out the specific criteria separated by “;” or I would put them in a simple table.

Results

10. On page 9, in the first paragraph in the Study characteristics section, I would add a sentence clarifying where the five RCTs were done – how many in the USA how many in the four outside of the USA (e.g., in more resource-constrained environments)?

11. On page 13, in the paragraph under the sub-heading “Summary of included studies,” the authors describe the adherence outcome and the fact that some studies used self-report and others used MEMS. This is a really important distinction because most ART adherence researchers accept that self-report is not a very accurate measure of adherence. So one recommendation is to specify clearly how many studies used self-report (Table 2 indicates that two did) here. Second, I think it will be important to acknowledge this limitation in the meta-analysis if studies that use self-reported adherence are included.

12. Table 2 – could you add the time frame for the adherence and viral load outcomes? It is important to know when those measures are taken. That is, for adherence, is it mean adherence over the intervention period, or in the last month? It’s hard to have a sense of the rigor of these studies without knowing when data were collected/analyzed for these outcomes.

13. In Table 2, for the last study, were both MEMS and self-reported adherence used? If so, it might be better to use “and” there to be clear.

14. On page 18, line 250, the authors write, “Sub-group analysis of studies found no difference in mean ART adherence among adolescents who receive only text message (one way)….”. Do you mean that analysis found no significant effect on adherence from one-way text messaging?

Discussion

15. I would add as a study limitation that adherence was measured in different ways, including the limitation related to self-reported adherence.

16. Finally – while the writing is generally clear, there are a number of minor grammar errors that need correcting throughout the manuscript.

6. PLOS authors have the option to publish the peer review history of their article (what does this mean?). If published, this will include your full peer review and any attached files.

Reviewer #1: **Yes: **Eric Green

Reviewer #2: No

---

## [Author Response · Author response to Decision Letter 0]

21 Mar 2021

We sincerely thank the reviewers for thorough reading of this manuscript and the editor for the opportunity to improve the manuscript. The suggestions and comments have been closely followed and revisions have been made accordingly. We have hopefully addressed the concerns to your satisfaction. Point-by-point responses to the comments are listed below. The revisions have been color tracked in the manuscript.

Reviewer #1: The authors stated that "the data are fully available without restriction" and "all relevant data are within the manuscript and its supporting files". I was not able to find any data files or analysis code. I'd be happy to complete a full review once the data and code are shared. I'm marking as "reject" at this point on the recommendation of the handling editor as a way to request access to these materials. This recommendation is not based on any assessment that the paper is unsuitable for publication.

Answer: We deeply regret the inconvenience to our reviewer. We have attached data files in the Appendix section: S3 Appendix – Data for meta-analysis. We are looking forward to receiving feedback from our reviewer.

Reviewer #2: To the authors:

This manuscript addresses an extremely important topic and I applaud the authors for undertaking this systematic review and meta-analysis to help us understand whether mobile phone SMS hold promise to support improved adherence in a known vulnerable population: adolescents with HIV. I enjoyed reading the manuscript. I found a few places that need clarification. With some careful editing, I believe it will be worthy of publication.

Specific Comments:

Abstract

1. In the Results section, 3rd sentence, the authors write, “… while no significant difference was found in the standard of care in the remaining two studies.” This is confusing. Do you mean that no difference was found between intervention and control (standard of care) groups? This should be clarified.

Answer: Thank you for pointing out this oversight. We understand that our statement was not clear at this point therefore, we have updated the sentence (Page number 2, line numbers 42 – 44) as follows: “Five studies showed a positive impact of text messaging in improving adherence, while no significant difference was found between the intervention and the control (standard of care) group in the remaining two studies.”

2. In the Conclusion and recommendation section, I think it would be good to highlight more prominently that this study could not confirm its motivating question: whether mobile phone text message interventions can improve adherence among HIV-positive adolescents. The meta-analysis failed to show a significant difference – it’s there in this section but it’s a bit buried between other points. This fundamental finding should be highlighted, I think.

Answer: As suggested by the reviewer, the Conclusion and recommendation section has been revised and it read as follows. (Page 2, line numbers 48 -54) “The meta-analysis did not show a statistically significant difference in improvement of ART adherence due to text message reminder intervention among adolescents living with HIV. The included studies were heterogeneous in the reported clinical outcomes where effectiveness of the intervention was identified in small studies which had a short follow-up period. Studies with a bigger sample size and a longer follow-up period are needed.”

Introduction

3. On page 3, second paragraph, there are some confusing numbers provided. In line 6 of the paragraph (lines 67-68) beginning with “With an emphasis on viral suppression…”, the sentence that says, “… of which 53% were aged 0-14 years and 68% aged 15 and older living with HIV” doesn’t make sense. Can you clarify these numbers?

Answer: Thank you for pointing out this oversight. The statement (Page number 4, line number 72-76) has been updated as follows: “At the end of 2019, 12.6 million (33% of people living with HIV) people did not have access to antiretroviral therapy (ART). An estimated 840 000 children did not have access to ART at the end of 2019, amounting to 47% of all children living with HIV.”

4. In the same paragraph, some of the references seem a bit old. For instance, the UNAIDS Global AIDS update is cited from 2017 – but the 2020 report is out, so I would recommend using the most recent numbers throughout this important background paragraph.

Answer: Thanks for this valuable input. Latest references have been updated on page 4, line numbers 74 and 75.

5. On page 4, in the first paragraph, top line of the page (line 75), the authors write, “Adherence to treatment, defined as how medication-taking behavior corresponds to the recommendations….” is not very helpful as an operational definition. What you mean is more like – “defined as taking HIV medications as prescribed, which includes taking them every day at roughly the same time, without missing doses.”

Answer: We have updated the definition of Adherence (page number 5, line numbers 82-85) as follows: “Adherence to treatment is defined as how patient’s medication-taking behaviour corresponds to their health care professional’s recommendations which include taking medications daily at about roughly the same time without missing doses.” 

6. For the last sentence in the same paragraph (line 81), why is the Finitsis article on text message interventions the reference for a general statement on the importance of adherence in preventing progression to AIDS, morbidity, and death among PLHIV? I think it would be good to have a reference to a study whose purpose was to investigate the relationship between adherence and morbidity/death (such as Paterson et al (2000) and Arnsten et al (2001)).

Answer: We agree with the reviewer on this point. We have changed our reference to Paterson et al (2000). (Page number 5, line number 90). 

7. On the next page (5), in the last paragraph (line 109), the authors write about their study focusing on the effect of text message reminders to improve ART adherence – but don’t specify that they are also focusing on a specific population: adolescents. This is an important part of their study and should be included here.

Answer: Thanks for pointing this out. We have updated the statement (Page number 6, line number 118-119) as “In this study the effect of a text message reminder intervention on ART adherence in adolescents living with HIV is explored by performing a systematic review and a meta-analysis.”

Methods

8. In the section on Study inclusion criteria, the text on the adherence measure seems confusing and inaccurate. The text reads (in line 138) “Adherence to ART either as pill count or viral load as an outcome.” One problem with this as written is that, while both outcomes are important, and HIV viral load can be an indicator of past adherence, it is not a measure of adherence itself, strictly speaking. The second problem is that saying “…ART either as pill count or viral load” suggests that adherence will be measured as pill count only. Pill count has a specific meaning in measuring adherence, and it is distinct from such measures as self-report, MEMS, etc. and the latter are clearly included in the papers included in the review – so this is very confusing. I would recommend dropping the “pill count” altogether (as that is misleading) and then revising the line somewhat to read: “using either adherence to ART (measured in various ways) or HIV viral load as a study outcome”. In reading the manuscript, this captures what the authors actually did.

Answer: We agree with the reviewer’s suggestion and have updated the statement (Page number 8, line numbers 148-149) as “Using either adherence to ART (measured in different ways) or viral load as the study outcome.”

9. For study inclusion/exclusion generally, I would not use bullets. I would recommend writing out the specific criteria separated by “;” or I would put them in a simple table.

Answer: Based on reviewer’s suggestion we have edited the text. (Page numbers 7-8, line numbers 142-158)

Results

10. On page 9, in the first paragraph in the Study characteristics section, I would add a sentence clarifying where the five RCTs were done – how many in the USA how many in the four outside of the USA (e.g., in more resource-constrained environments)?

Answer: We have updated the text (Page number 10, line numbers 206-209) and it reads as follows: “Seven studies were included in the review; five RCTs (two in the USA and one each in Uganda, Nigeria, and Cameroon) and two cohort studies (one each in USA and Argentina) analysing the effect of mobile text reminders for improving adherence to ART among adolescents living with HIV were included”

11. On page 13, in the paragraph under the sub-heading “Summary of included studies,” the authors describe the adherence outcome and the fact that some studies used self-report and others used MEMS. This is a really important distinction because most ART adherence researchers accept that self-report is not a very accurate measure of adherence. So one recommendation is to specify clearly how many studies used self-report (Table 2 indicates that two did) here. Second, I think it will be important to acknowledge this limitation in the meta-analysis if studies that use self-reported adherence are included.

Answer: Thanks for this valuable insight. We have revised the text (page numbers 14-15, line numbers 233-237) as “Three studies (Hailey 2013, Abiodun 2019 and Ketchaji 2019) used self-reported data for outcome assessment, while two studies (Linnemayr 2014 and Mimiaga 2019) documented using a medication event monitoring system (MEMS) in addition to self-reported data.”.

As suggested by the reviewer we have updated our limitations of meta-analysis (page number 23, line numbers 342-345) as follows: “Moreover, five studies used self-reported data, which is considered a less sensitive measure of adherence than MEMS and tends to overestimate the impact. We think that inclusion of self-reported data in our meta-analysis may affect the validity of our findings and therefore is a limitation of this study.”

12. Table 2 – could you add the time frame for the adherence and viral load outcomes? It is important to know when those measures are taken. That is, for adherence, is it mean adherence over the intervention period, or in the last month? It’s hard to have a sense of the rigor of these studies without knowing when data were collected/analyzed for these outcomes.

Answer: We duly acknowledge reviewer’s views on the issue. We have changed the heading of the last column in Table 2 (page number 16) to “Adherence and/or viral load measurement1”. We have further added a note at the bottom of the table (page number 18) “Adherence and viral load measured at baseline and at the end of the follow-up period.” We hope that our modification can address the issue raised by the reviewer. 

13. In Table 2, for the last study, were both MEMS and self-reported adherence used? If so, it might be better to use “and” there to be clear.

Answer: Thanks for the suggestion. We have updated Table 2 (page number 18) in this regard.

14. On page 18, line 250, the authors write, “Sub-group analysis of studies found no difference in mean ART adherence among adolescents who receive only text message (one way)….”. Do you mean that analysis found no significant effect on adherence from one-way text messaging?

Answer: At this point we wanted to say that the mean difference between one-way text messaging and control is zero. We have revised are statement (page number 19, line numbers 250-252) as “Sub-group analysis of studies found no difference in mean ART adherence (mean difference = 0.00) among adolescents who receive only text message (one way).” to clarify our viewpoint.

Discussion

15. I would add as a study limitation that adherence was measured in different ways, including the limitation related to self-reported adherence.

Answer: Thank you for this valuable insight. We have updated our limitations of meta-analysis (page number 23, line numbers 342-345) as follows: “Moreover, five studies used self-reported data, which is considered a less sensitive measure of adherence than MEMS and tends to overestimate the impact. We think that inclusion of self-reported data in our meta-analysis may affect the validity of our findings and therefore is a limitation of this study.”

16. Finally – while the writing is generally clear, there are a number of minor grammar errors that need correcting throughout the manuscript.

Answer: We have tried to correct grammatical errors in the manuscript. Additionally, we had our manuscript proofread by a native English speaker who has experience in academic writing.

---

## [Decision Letter · Decision Letter 1]

14 Apr 2021

PONE-D-21-02241R1

Effectiveness of mobile phone text message reminder interventions to improve adherence to antiretroviral therapy among adolescents living with HIV: a systematic review and meta-analysis.

PLOS ONE

Dear Dr. Mehra,

Thank you for submitting your manuscript to PLOS ONE. After careful consideration, we feel that it has merit but does not fully meet PLOS ONE’s publication criteria as it currently stands. Therefore, we invite you to submit a revised version of the manuscript that addresses the points raised during the review process.

We look forward to receiving your revised manuscript.

Kind regards,

Giuseppe Vittorio De Socio, MD, PhD

Academic Editor

PLOS ONE

Reviewers' comments:

Reviewer's Responses to Questions

**Comments to the Author**

1. If the authors have adequately addressed your comments raised in a previous round of review and you feel that this manuscript is now acceptable for publication, you may indicate that here to bypass the “Comments to the Author” section, enter your conflict of interest statement in the “Confidential to Editor” section, and submit your "Accept" recommendation.

Reviewer #1: (No Response)

Reviewer #2: (No Response)

2. Is the manuscript technically sound, and do the data support the conclusions?

Reviewer #1: Partly

Reviewer #2: Yes

3. Has the statistical analysis been performed appropriately and rigorously? 

Reviewer #1: No

Reviewer #2: Yes

4. Have the authors made all data underlying the findings in their manuscript fully available?

Reviewer #1: Yes

Reviewer #2: Yes

5. Is the manuscript presented in an intelligible fashion and written in standard English?

Reviewer #1: Yes

Reviewer #2: No

6. Review Comments to the Author

Reviewer #1: The authors conducted a meta-analysis of SMS reminder interventions on ART adherence among HIV-positive adolescents. My main concern is that the outcome is not clearly defined, and this made it hard for me to interpret the findings.

The authors write:

- "Adherence to treatment is defined as how patient’s medication-taking behaviour corresponds to their health care professional’s recommendations which include taking medications daily at about roughly the same time without missing doses."

- "adherence to ART (measured in different ways)"

- "The data used in the analysis were mean adherence rates"

- "Medication adherence was reported as a mean adherence rate, odds ratio, or using a visual analogue scale (VAS) score."

Is adherence a binary outcome? If so, what made someone "adherent" in the primary studies?

Is adherence defined as somone's adherent days / all days during each trial? Is it a pill count?

I searched a bit and confirmed my hunch that many (most?) meta-analyses of medication adherence studies report the effect sizes as odds ratios. It's possible that I'm out of touch on this, but I would benefit from an explanation from the authors about why mean differences of rates makes the most sense for this meta-analysis.

My other comments are minor and should not stand in the way of publication if that is the editor's decision:

- Unclear why cohort studies were included

- Unclear why interventions were limited to SMS (the field has expanded so much since SMS was dominant, e.g., WhatsApp)

- The introduction does not mention anything about the evidence in favor of SMS reminders for adherence among adults living with HIV (does appear in the discussion)

- The overall reporting of the methods and results could be improved

- Why is the age range of adolescents limited to 19? The definition varies across settings. I think more studies could have been included with a broader definition.

Reviewer #2: In this resubmitted version of the manuscript, the authors addressed the points in my original review. Some areas of the revised text are still a bit confusing, however. With a few relatively minor edits, I believe the manuscript will be worthy of publication and make an important contribution to the field.

Specific Comments:

Abstract

1. The authors clarified the main study finding, that the meta-analysis failed to show a significant difference of mobile phone text message reminders, but the text is a bit confusing (Page 2, line numbers 48-49). I would reorganize the first sentence slightly for clarity as follows: “The meta-analysis of text message reminder interventions did not show a statistically significant difference in improvement of ART adherence among adolescents living with HIV.”

Introduction

2. With reference to the edits made to define adherence (page number 4, line numbers 78-81), the authors made helpful edits, but “about” should be removed in the following sentence to avoid redundancy: “Adherence to treatment is defined as how patient’s medication-taking behaviour corresponds to their health care professional’s recommendations which include taking medications daily at [about—remove] roughly the same time without missing doses.”

Discussion

3. Adding that use of self-report as an adherence measure by some of the included studies in the meta-analysis is a study limitation is helpful (page 22, lines 342-345). However, some of that text is not quite accurate, as use of self-report overstates adherence, not impact. Rather than writing “…tends to overestimate the impact” I suggest that the authors say “…tends to overestimate adherence.” Also, it would be good to introduce that paragraph on limitations with something like: “We also acknowledge a number of study limitations…..”

4. Finally – there are still a lot of minor grammar errors and typos that need correcting in the manuscript. I'm sorry I don't have time to point out all the places that need editing. However, here are two examples, both from the Methods section of the Abstract. First, the second sentence beginning with “Literature search was done…” should read, “A literature search was done…” Second, the next to last sentence which ends with, “…computed using a random effect model” should read “…computed using a random effects model." The manuscript needs careful editing by a native English speaker to remove all the errors and typos before it can be published.

7. PLOS authors have the option to publish the peer review history of their article (what does this mean?). If published, this will include your full peer review and any attached files.

Reviewer #1: **Yes: **Eric Green

Reviewer #2: No

---

## [Author Response · Author response to Decision Letter 1]

22 May 2021

Response to reviewers

We sincerely thank the reviewers for the thorough reading of this manuscript and the editor for the opportunity to improve the manuscript. The suggestions and comments have been closely followed, and revisions have been made accordingly. We have hopefully addressed the concerns to your satisfaction. Point-by-point responses to the comments are listed below. In addition, the revisions have been colour tracked in the manuscript.

Reviewer #1: The authors conducted a meta-analysis of SMS reminder interventions on ART adherence among HIV-positive adolescents. My main concern is that the outcome is not clearly defined, and this made it hard for me to interpret the findings.

The authors write:

- "Adherence to treatment is defined as how patient’s medication-taking behaviour corresponds to their health care professional’s recommendations which include taking medications daily at about roughly the same time without missing doses."

- "adherence to ART (measured in different ways)"

- "The data used in the analysis were mean adherence rates"

- "Medication adherence was reported as a mean adherence rate, odds ratio, or using a visual analogue scale (VAS) score."

Is adherence a binary outcome? If so, what made someone "adherent" in the primary studies?

Response: Thank you for raising this important question. There was considerable heterogeneity in how adherence was reported in primary studies. Four out of seven primary studies reported adherence as a continuous variable where intervention was compared with a control group (Linnemayr et. al., Abiodun et al and Mimiaga et. al.) or a pre-and post-intervention adherence level was reported (Hailey et al). In these studies, the participants receiving the intervention were reported to be more (or less) adherent as compared to a control group and/or a pre-intervention state.

The study by Abiodun et. al., additionally reported a dichotomised variable based on a VAS score of >= 95%. Study by Ketchaji et. al., also reported a binary variable for adherence where criteria for dichotomisation was not reported. 

The heterogeneity of the outcomes (continuous or binary) reported in the primary studies affected our choice of outcome measure for meta-analysis. Primary studies dichotomised mean adherence at different points. Therefore, we have updated our limitations as follows (Page 22, lines 345-347): “Finally, variation in reported outcome (continuous or binary) and their units posed a challenge to include most studies in the meta-analysis.”

Is adherence defined as somone's adherent days / all days during each trial? Is it a pill count?

Response: Thanks for highlighting this oversight. In the present study, adherence as an outcome variable is measured either as a pill count or MEMS counts (prescription drug bottle opening counts). In this regard, we have added the following sentences to our methodology (Page 8, lines 182-187) “Most studies included in this systematic review reported adherence to ART as a continuous outcome (mean adherence). Therefore, mean adherence was chosen as the outcome measure for quantitative analysis. For the meta-analysis, adherence is defined as how a participants’ pill count or MEMS data corresponds to their doctor’s prescription regimen during the period of trial. Mean adherence is a continuous variable that can take a value from 0-1, where 0 represents nil adherence, and 1 represent perfect adherence.”.

I searched a bit and confirmed my hunch that many (most?) meta-analyses of medication adherence studies report the effect sizes as odds ratios. It's possible that I'm out of touch on this, but I would benefit from an explanation from the authors about why mean differences of rates makes the most sense for this meta-analysis.

Response: We thank the reviewer for asking this important question. We agree with the reviewer that most studies used the odds ratio. Mean adherence is an absolute measure of effect as opposed to odds ratio, which is a relative measure of effect. The outcome variable needs to be binary to investigate the odds ratio. If the outcome measure for the analysis is a continuous variable, then it needs to be dichotomised at a specific point. For ART adherence, mean adherence rate is dichotomised at the point where viral suppression is assumed to have achieved.

In the current study, we undertook a comprehensive review of peer-reviewed and grey literature. We decided that having an absolute measure of effect (mean difference) in our meta-analysis will be more suitable due to the following reasons:

1. Using mean adherence rates allowed us to include more studies in the meta-analysis.

2. We were interested in examining whether text message reminder can improve adherence. Mean difference is a commonly used measure of effect in meta-analysis.

3. Mean difference evaluates a continuous variable (mean adherence) and, therefore, is not based on the assumption that viral suppression is achieved at a specific level of adherence. 

In this regard, we have updated our methods section as follows (Page 8, lines 182-184) "Most included studies in this systematic review reported adherence to ART as a continuous outcome (mean adherence). Therefore, mean adherence was chosen as the outcome measure for quantitative analysis.". We further updated our limitations as (Page 22, lines 347-350) "Odds ratio is a more common outcome measure for adherence in a meta-analysis that evaluates a binary outcome. However, in the current study, the included studies dichotomised adherence at different points that posed a challenge to have odds ratio as an outcome measure."

My other comments are minor and should not stand in the way of publication if that is the editor's decision:

- Unclear why cohort studies were included

Response: We thank the reviewer for raising this question. The current study aimed to gather evidence of the effectiveness of text message reminder intervention. In addition to randomised studies, a cohort (non-randomised) study design can demonstrate the effectiveness of interventions. Therefore, cohort studies were made a part of the inclusion criteria. The inclusion criteria were decided a priori, given the limited number of randomised trials anticipated in this area.

- Unclear why interventions were limited to SMS (the field has expanded so much since SMS was dominant, e.g., WhatsApp)

Response: We thank the reviewer for this comment. The accessibility of smartphones (having WhatsApp) has been increasing worldwide, especially in high and upper-middle income countries but is currently lower in low and low-middle income countries. So, we first wanted to determine if this primary digital function, SMS, which is available almost everywhere, could have any beneficial impact. In the "Introduction", we have the following statement (Page 5, lines 120-122) "A text message can be received on a mobile phone with basic features. This intervention curtails stigma and can be rolled out in a resource-limited setting.".

- The introduction does not mention anything about the evidence in favor of SMS reminders for adherence among adults living with HIV (does appear in the discussion)

Response: We thank the reviewer for this question. We would like to mention that in the “Introduction” section, we have the following statement to inform on the evidence in favour of SMS reminder for adherence among adults living with HIV (Page 5-6, lines 121-124) “Previous studies have shown the effectiveness of mHealth interventions in improving treatment adherence among adults living with HIV (35-38), and the evidence is emerging in the adolescent age group.”. Four peer-reviewed references support our statement. We acknowledge that we have a more detailed narration on these studies in our "Discussion" section, where we have compared our findings with previous studies in this area.

- The overall reporting of the methods and results could be improved

Response: We thank the reviewer for this helpful suggestion. Based on the reviewers' comments and suggestions, we have tried to improve our methods and results section and tracked changes are made in the manuscript in this regard.

- Why is the age range of adolescents limited to 19? The definition varies across settings. I think more studies could have been included with a broader definition.

Response: We agree more studies could have been included if the age criteria was expanded. However, we utilised the definition of adolescents age group as recommended by the World Health Organisation. We aimed to focus our evidence search on the adolescent age group (defined by WHO) for two reasons. Firstly, we found out some unique challenges related to ART adherence and retention care in this age group. Secondly, we wanted to focus on identifying adolescent-friendly interventions that can improve adherence and retention in care in this unique age group. 

Reviewer #2: In this resubmitted version of the manuscript, the authors addressed the points in my original review. Some areas of the revised text are still a bit confusing, however. With a few relatively minor edits, I believe the manuscript will be worthy of publication and make an important contribution to the field.

Specific Comments:

Abstract

1. The authors clarified the main study finding, that the meta-analysis failed to show a significant difference of mobile phone text message reminders, but the text is a bit confusing (Page 2, line numbers 48-49). I would reorganize the first sentence slightly for clarity as follows: “The meta-analysis of text message reminder interventions did not show a statistically significant difference in improvement of ART adherence among adolescents living with HIV.”

Response: Thanks for the suggestion. We have rewritten the statement as advised (Page 2, line number 48-50).

Introduction

2. With reference to the edits made to define adherence (page number 4, line numbers 78-81), the authors made helpful edits, but “about” should be removed in the following sentence to avoid redundancy: “Adherence to treatment is defined as how patient’s medication-taking behaviour corresponds to their health care professional’s recommendations which include taking medications daily at [about—remove] roughly the same time without missing doses.”

Response: Thanks for the suggestion, we have rewritten the statement as advised (Page 4, line number 78-81).

Discussion

3. Adding that use of self-report as an adherence measure by some of the included studies in the meta-analysis is a study limitation is helpful (page 22, lines 342-344). However, some of that text is not quite accurate, as use of self-report overstates adherence, not impact. Rather than writing “…tends to overestimate the impact” I suggest that the authors say “…tends to overestimate adherence.” Also, it would be good to introduce that paragraph on limitations with something like: “We also acknowledge a number of study limitations…..”

Response: Thanks for this valuable suggestion. We have added a sentence (Page 22, lines 339) “We also acknowledge some limitations in our study.” 

We have rewritten another statement in the paragraph (Page 22, lines 342-344) “Moreover, five studies used self-reported data, which is considered a less sensitive measure of adherence than MEMS and tends to overestimate the adherence”.

4. Finally – there are still a lot of minor grammar errors and typos that need correcting in the manuscript. I'm sorry I don't have time to point out all the places that need editing. However, here are two examples, both from the Methods section of the Abstract. First, the second sentence beginning with “Literature search was done…” should read, “A literature search was done…” Second, the next to last sentence which ends with, “…computed using a random effect model” should read “…computed using a random effects model." The manuscript needs careful editing by a native English speaker to remove all the errors and typos before it can be published.

Response: Thanks for pointing this out. A native English speaker has reviewed the manuscript. We have made tracked changes throughout the manuscript.

---

## [Decision Letter · Decision Letter 2]

17 Jun 2021

PONE-D-21-02241R2

Effectiveness of mobile phone text message reminder interventions to improve adherence to antiretroviral therapy among adolescents living with HIV: A systematic review and meta-analysis.

PLOS ONE

Dear Dr. Mehra,

Thank you for submitting your manuscript to PLOS ONE. After careful consideration, we feel that it has merit but does not fully meet PLOS ONE’s publication criteria as it currently stands. Therefore, we invite you to submit a revised version of the manuscript that addresses the points raised during the review process.

See comment by Reviewer #2.

We look forward to receiving your revised manuscript.

Kind regards,

Giuseppe Vittorio De Socio, MD, PhD

Academic Editor

PLOS ONE

Journal Requirements:

Reviewers' comments:

Reviewer's Responses to Questions

**Comments to the Author**

1. If the authors have adequately addressed your comments raised in a previous round of review and you feel that this manuscript is now acceptable for publication, you may indicate that here to bypass the “Comments to the Author” section, enter your conflict of interest statement in the “Confidential to Editor” section, and submit your "Accept" recommendation.

Reviewer #1: All comments have been addressed

Reviewer #2: (No Response)

2. Is the manuscript technically sound, and do the data support the conclusions?

Reviewer #1: (No Response)

Reviewer #2: Yes

3. Has the statistical analysis been performed appropriately and rigorously? 

Reviewer #1: (No Response)

Reviewer #2: Yes

4. Have the authors made all data underlying the findings in their manuscript fully available?

Reviewer #1: (No Response)

Reviewer #2: Yes

5. Is the manuscript presented in an intelligible fashion and written in standard English?

Reviewer #1: (No Response)

Reviewer #2: No

6. Review Comments to the Author

Reviewer #1: (No Response)

Reviewer #2: Thank you for addressing the comments of reviewers of the previous submission. It is much improved since the previous version. I have two remaining issues that I hope the authors will address.

1. On page 4, line 83, the authors write that adherence >95% is necessary for optimal viral suppression, and cite an article from 2000. ART regimens have improved substantially since 2000 and the precise level of “necessary” adherence for viral suppression is unclear (and is somewhat of a controversial subject). Numerous articles since 2000 have highlighted this point. Please see, for example:

**Bangsberg DR. Less than 95% adherence to nonnucleoside reverse-transcriptase inhibitor therapy can lead to viral suppression. Clin Infect Dis. Oct 1 2006; 43(7): 939-41.

**Parienti JJ, Das-Douglas M, Massari V, Guzman D, Deeks SG, Verdon R, Bangsberg DR. Not all missed doses are the same: sustained NNRTI treatment interruptions predict HIV rebound at low-to-moderate adherence levels. PLoS One. 2008; 3(7): e2783.

**Haberer JE, Sabin L, Amico KR, Orrell C, Galarraga O, Tsai AC, et al. Improving antiretroviral therapy adherence in resource-limited settings at scale: a discussion of interventions and recommendations. J Int AIDS Soc. 2017; 20(1): 21371.

2. The authors state that a native English speaker has reviewed the manuscript, but it still has numerous typos and sentences with incorrect grammar. Two examples are from the first page of the main manuscript: 1) Line 60: the sentence now beginning with “Latest estimates suggest…” should start with “The” so that the sentence reads, “The latest estimates suggest….”; 2) Lines 69-70: the sentence that reads “At the end of 2019, 12.6 million (33% of people living with HIV) people did not have…” should be revised to read: “At the end of 2019, 12.6 million people (33% of people living with HIV) did not have….” I suggest that the authors can do further proofreading and editing so that the manuscript is error free.

7. PLOS authors have the option to publish the peer review history of their article (what does this mean?). If published, this will include your full peer review and any attached files.

Reviewer #1: **Yes: **Eric Green

Reviewer #2: No

---

## [Author Response · Author response to Decision Letter 2]

4 Jul 2021

We sincerely thank the reviewers for the thorough reading of this manuscript and the editor for the opportunity to improve the manuscript. The suggestions and comments have been closely followed, and revisions have been made accordingly. We have hopefully addressed the concerns to your satisfaction. Point-by-point responses to the comments are listed below. In addition, the revisions have been colour tracked in the manuscript.

Reviewer #2: Thank you for addressing the comments of reviewers of the previous submission. It is much improved since the previous version. I have two remaining issues that I hope the authors will address.

1. On page 4, line 83, the authors write that adherence >95% is necessary for optimal viral suppression, and cite an article from 2000. ART regimens have improved substantially since 2000 and the precise level of “necessary” adherence for viral suppression is unclear (and is somewhat of a controversial subject). Numerous articles since 2000 have highlighted this point. Please see, for example:

**Bangsberg DR. Less than 95% adherence to nonnucleoside reverse-transcriptase inhibitor therapy can lead to viral suppression. Clin Infect Dis. Oct 1 2006; 43(7): 939-41.

**Parienti JJ, Das-Douglas M, Massari V, Guzman D, Deeks SG, Verdon R, Bangsberg DR. Not all missed doses are the same: sustained NNRTI treatment interruptions predict HIV rebound at low-to-moderate adherence levels. PLoS One. 2008; 3(7): e2783.

**Haberer JE, Sabin L, Amico KR, Orrell C, Galarraga O, Tsai AC, et al. Improving antiretroviral therapy adherence in resource-limited settings at scale: a discussion of interventions and recommendations. J Int AIDS Soc. 2017; 20(1): 21371.

Response: We would like to thank our reviewer for this valuable insight. We have updated our text and included relevant references. The following statements have been included in the manuscript on page 4 Line 84-86 – “However, recent studies suggest that even moderate adherence to potent ART regimens can achieve viral suppression (Bangsberg, 2006). Parienti et al. point to sustained treatment interruptions leading to viral rebound rather than interspersed missed doses (Parienti, 2006).”.

2. The authors state that a native English speaker has reviewed the manuscript, but it still has numerous typos and sentences with incorrect grammar. Two examples are from the first page of the main manuscript: 1) Line 60: the sentence now beginning with “Latest estimates suggest…” should start with “The” so that the sentence reads, “The latest estimates suggest….”; 2) Lines 69-70: the sentence that reads “At the end of 2019, 12.6 million (33% of people living with HIV) people did not have…” should be revised to read: “At the end of 2019, 12.6 million people (33% of people living with HIV) did not have….” I suggest that the authors can do further proofreading and editing so that the manuscript is error free.

Response: We appreciate our reviewer’s concern. To ensure that our manuscript is error free, it has been reviewed again by a native English speaker. The grammatical corrections are shown as tracked changes throughout the manuscript.

Changes to the reference List 

We had one retracted refence in our list that was replaced with a new reference. Furthermore, we identified that four of references are available at a different web location and therefore, they were updated. Lastly, we included 2 new references as per our reviewer’s suggestion above.

Retracted reference – (6) United Nations Children's Fund. Early and late adolescence: UNICEF; 2011 [Available from: https://www.unicef.org/sowc2011/pdfs/Early-and-late-adolescence.pdf].

New reference – (6) World Health Organization. Stages of Adolescent Development 2010 [Available from: https://apps.who.int/adolescent/second-decade/section/section_2/level2_2.php].

Old reference – (10) World Health Organization. Adherence to long-term therapies: evidence for action: World Health Organization; 2003 [Available from: https://www.who.int/chp/knowledge/publications/adherence_report/en/].

Updated reference – (10) World Health Organization. Adherence to long-term therapies: evidence for action Switzerland2003 [Available from: https://apps.who.int/iris/bitstream/handle/10665/42682/9241545992.pdf].

Old reference – (19) World Health Organization. Chapter 9: Guidance on operations and service delivery: WHO; 2013 [Available from: https://www.who.int/hiv/pub/guidelines/arv2013/operational/adherence/en/].

Updated reference – (19) World Health Organization. Chapter 9: Guidance on operations and service delivery: WHO; 2013 [Available from: https://www.who.int/publications/i/item/9789241505727].

Old reference – (23) World Health Organization. HIV and adolescents: guidance for HIV testing and counselling and care for adolescents living with HIV: recommendations for a public health approach and considerations for policy-makers and managers 2013 [Available from: https://www.who.int/hiv/pub/guidelines/adolescents/en/].

Updated reference – (23) World Health Organization. HIV and adolescents: guidance for HIV testing and counselling and care for adolescents living with HIV: recommendations for a public health approach and considerations for policy-makers and managers 2013 [Available from: https://www.who.int/publications/i/item/9789241506168].

Old reference – (26) World Health Organization. Global diffusion of eHealth: Making universal health coverage achievable. Report of the third global survey on eHealth: WHO; 2016 [Available from: https://www.who.int/goe/publications/global_diffusion/en/].

Updated reference – (26) World Health Organization. Global diffusion of eHealth: Making universal health coverage achievable. Report of the third global survey on eHealth: WHO; 2016 [Available from: https://www.who.int/publications/i/item/9789241511780].

Added reference – (14) Bangsberg D. R. Less Than 95% Adherence to Nonnucleoside Reverse-Transcriptase Inhibitor Therapy Can Lead to Viral Suppression. Clinical Infectious Diseases. 2006;43(7):939-41. doi 10.1086/507526

Added reference – (15) Parienti J. J., Das-Douglas M., Massari V., Guzman D., Deeks S. G., Verdon R. et al. Not all missed doses are the same: sustained NNRTI treatment interruptions predict HIV rebound at low-to-moderate adherence levels. PLoS One. 2008;3(7):e2783. doi 10.1371/journal.pone.0002783

---

## [Editor Report · Decision Letter 3]

7 Jul 2021

Effectiveness of mobile phone text message reminder interventions to improve adherence to antiretroviral therapy among adolescents living with HIV: A systematic review and meta-analysis.

PONE-D-21-02241R3

Dear Dr. Mehra,

We’re pleased to inform you that your manuscript has been judged scientifically suitable for publication and will be formally accepted for publication once it meets all outstanding technical requirements.

Kind regards,

Giuseppe Vittorio De Socio, MD, PhD

Academic Editor

PLOS ONE
---

## [Editor Report · Acceptance letter]

12 Jul 2021

PONE-D-21-02241R3 

Effectiveness of mobile phone text message reminder interventions to improve adherence to antiretroviral therapy among adolescents living with HIV: A systematic review and meta-analysis. 

Dear Dr. Mehra:

I'm pleased to inform you that your manuscript has been deemed suitable for publication in PLOS ONE. Congratulations! Your manuscript is now with our production department. 

Kind regards, 

on behalf of

Dr. Giuseppe Vittorio De Socio 

Academic Editor

PLOS ONE